# Anti-Acne Vulgaris Effects of Pedunculagin from the Leaves of *Quercus mongolica* by Anti-Inflammatory Activity and 5α-Reductase Inhibition

**DOI:** 10.3390/molecules25092154

**Published:** 2020-05-05

**Authors:** Min Kim, Jun Yin, In Hyeok Hwang, Dong Hui Park, Eun Kyeong Lee, Min Ji Kim, Min Won Lee

**Affiliations:** Department of Pharmacognosy and Natural product-derived Medicine, College of Pharmacy, Chung-Ang University, Seoul 156–756, Korea; km9477@naver.com (M.K.); yinjun89@naver.com (J.Y.); grampus92@naver.com (I.H.H.); donghee9611@naver.com (D.H.P.); xkfanem09@naver.com (E.K.L.); kam4256@naver.com (M.J.K.)

**Keywords:** *Quercus mongolica*, pedunculagin, anti-acne vulgaris

## Abstract

*Quercus mongolica* (QM)—a member of the Fagaceae family—has been used as traditional medicine in Korea, China and Mongolia as a treatment for inflammation of oral, genital or anal mucosa and for external inflammation of skin. To treat acne vulgaris (AV), we evaluated the inhibition of inflammatory cytokines (IL-6 and IL-8) of QM leaf extract (QML) and its main compound, pedunculagin (PD) in vitro and 5α-reductase inhibitory activity by western blotting. As results, QML and PD showed potent NO production inhibitory activity compared with the positive control (PC), NG-monomethyl-L-arginine (L-NMMA). QML and PD was also showed the decreases of IL-6 and IL-8 compared with the PC, EGCG and exhibited potent 5α-reductase type 1 inhibitory activities compared with the PC, dutasteride.

## 1. Introduction

Keratinocytes play a role in skin barrier function. When these keratinocytes are exposed to external stimuli like air pollution, stress and ultraviolet radiations it can induce mediator of inflammation, nitric oxide (NO) and inflammatory response such as inflammatory cytokines. Hence, NO and the inflammatory cytokines may cause acne vulgaris (AV) [1,2,3,4].

NO is a free radical that damages to cell, tissue and DNA. And NO can form peroxynitrite that is a potent cytotoxic agent damage to molecules in cells, including DNA. These damages evoke many chronic inflammatory diseases such as psoriasis and pathogenesis of AV. It also was reported that nitrate and nitrite levels which are an index of NO production were high in acne patients [5,6].

Keratinocytes produce many confirmed cytokines containing interleukin-1 (IL-1), IL-6, IL-7, IL-8, IL-10, IL-12 [4,7]. Particularly, IL-6 induce differentiation of Th17 cells. Activation of the Th17 cell produce IL-17 that the cytokine contributes significantly to the inflammatory response in AV. In addition, IL-17 found in acne lesions [8,9,10]. And IL-8 is among CXC chemokines induced AV by attracting neutrophils to pilosebaceous unit [11,12].

AV is a chronic inflammatory disease caused by excessive sebum production [13,14]. Sebum production relates to androgen. Particularly, dihydrotestosterone (DHT) more potent triggers AV approximately 5–10 times by increasing excessive sebum production. And increased DHT may act on infundibular keratinocytes leading to abnormal hyperkeratinization. If abnormal hyperkeratinization occur in keratinocytes, sebum production increases excessively in sebaceous gland [15,16,17]. 5α-reductase converts DHT from testosterone. Thus, inhibition of 5α-reductase activity improves AV by reducing DHT [18]. Several studies reported that natural sources and its active components, mainly polyphenols improve AV by inhibition of 5α-reductase activity [19,20,21,22,23].

*Quercus mongolica* (QM) is a deciduous oak that, has been used in oriental traditional medicine in north east Asia. It was used in Korea, China and Japan for the treatment of the inflammation of oral, genital or anal mucosa and externally for the inflammation of skin [24,25]. Leaves of QM contains flavonoids, tannins, triterpenoids and phenols. These components has been reported to possess anti-oxidative, anti-inflammatory, antitumor, anti-microbial, anti-allergic and anti-fungal activities [26]. In particular, pedunculagin (PD) which isolated QM had an effect on potent inhibitory activities of chemokine and cytokine in keratinocytes. PD has also been reported to enhance the regeneration of keratinocytes [27,28].

In spite of many studies conducted on the various effects of QM, this study was conducted to find the improving agent of AV from QM leaf extract (QML) and PD.

## 2. Results

### 2.1. Isolation of PD

QM leaves (4.69 kg) pulverized and extracted with 80% acetone for 72 h, at room temperature to obtain QML (1.4 kg). Repeated column chromatography of QML and its subfraction (45.75 g) using Sephadex LH 20 gel to yield PD (1 g). The structure of the PD was identified by analysis of ^1^H-NMR and ^13^C-NMR spectra and comparison with reference [29].

^1^H NMR (600 MHz, Acetone-d_6_+D_2_O): δ α-glucose 5.40 (1/2H, d, *J* = 3.6 Hz, H-1α), 5.02 (1/2H, dd, *J* = 3.6, 10.0 Hz, H-2α), 5.42 (1/2H, t, *J* = 10.0 Hz, H-3α), 5.03 (1/2H, t, *J* = 10.0 Hz, H-4α), 4.56 (1/2H, ddd, *J* = 1.5, 6.6, 10.0 Hz, H-5α), 5.21 (1/2H, dd, *J* =6.9, 13.0 Hz, H-6aα), 3.85 (1/2H, dd, *J* =1.5, 13.0 Hz, H-6bα), β-glucose 5.02 (1/2H, d, *J* =7.8 Hz, H-1β), 4.82 (1/2H, dd, *J* =8.0, 9.0 Hz, H-2β), 5.18 (1/2H, dd, *J* =9.0, 10.0 Hz, H-3β), 5.04 (1/2H, t, *J* =10.0 Hz, H-4β), 4.17 (1/2H, ddd, *J* =0.9, 6.6, 10.0 Hz, H-5β), 5.25 (1/2H, dd, *J* =6.3, 13.0 Hz, H-6aβ), 3.78 (1/2H, dd, *J* =0.9, 13.0 Hz, H-6bβ).

^13^C NMR (600 MHz, Acetone-d_6_+D_2_O): δ 63.08 (G-6α), 63.10 (G-6β), 66.69 (HHDP-5α), 69.12 (G-4β), 69.47 (G-4α), 71.78 (HHDP-4β), 75.05 (HHDP-4α), 75.33 (HHDP-5β), 77.13 (HHDP-2α), 77.64 (HHDP-3α), 77.9 (HHDP-3β), 78.4 (HHDP-2β), 90.98 (G-1α), 94.63 (G-1β), 106.80 (HHDP-6α), 106.82 (HHDP-6β), 106.99 (HHDP-3′), 107.04 (HHDP-3), 107.11 (C-1), 107.61 (C-1′), 107.62 (C-1′’), 113.90 (C-2′’), 114.27 (C-2′), 115.32 (C-2), 125.14 (HHDP-2), 125.21 (HHDP-2′), 125.60 (HHDP-5), 125.65 (HHDP-5′), 125.8 (C-5′’), 125.82 (C-5′), 135.62 (C-5), 135.86 (C-4′’), 135.88 (C-4′), 143.78 (C-4), 143.79 (HHDP-4), 143.87 (HHDP-4′), 143.88 (HHDP-6), 143.94 (HHDP-6′), 144.57(C-3′), 144.59 (C-3′’), 144.70 (C-3), 167.79–169.25 (-COO).

### 2.2. Inhibitory Activity on NO Production

Inhibitory activity on NO production of QML and PC was measured to assess the anti-inflammatory activities in RAW 246.7 cells. QML (IC_50_ = 1.45 ± 0.25 μg/mL) showed potent anti-inflammatory activities compared with the positive control (PC), NG-monomethyl-L-arginine (L-NMMA) (IC_50_ = 0.55 ± 0.49 μg/mL). PD (IC_50_ = 53.52 ± 9.34 µM) adequately reduced NO production compared to L-NMMA (IC_50_ = 14.81 ± 12.76 μg/mL) (Table 1).

### 2.3. Cytotoxic Activity

Before assessing improvement effects on anti-AV, MTT assay was measured to assess the cytotoxic activity of QML and PD on RAW 264.7 cells and HaCaT cells. The cytotoxic activity of QML and PD was not observed at various concentrations (12.5, 25, 50 and 100 μg/mL or μM) (data not shown).

### 2.4. Inhibitory Activity on Cytokine Production

The LPS (1 μg/mL) causing the inflammation treated in HaCaT cells to evaluate the inhibitory effects of IL-6, IL-8 production. After exposure to LPS, the inhibitory activity on IL-6, IL-8 production of QML and PD was measured to assess the anti- inflammatory activities. The IL-6 concentration was decreased in the sample-treated groups. QML (IC_50_ = 9.37 ± 1.50 μg/mL) showed potent anti-inflammatory activities compared with the PC, EGCG (IC_50_ = 2.98 ± 1.47 μg/mL). PD (IC_50_ = 6.59 ± 1.66 µM) appeared stronger anti-inflammatory activities than EGCG (IC_50_ = 6.68 ± 1.86 μg/mL) (Table 2).

The IL-8 concentration was decreased in the sample-treated groups. QML (IC_50_ = 6.38 ± 2.58 μg/mL) showed potent anti-inflammatory activities compared with the PC, EGCG (IC_50_ = 0.74 ± 0.09 μg/mL). PD (IC_50_ = 0.09 ± 0.41 µM) appeared stronger anti-inflammatory activities than EGCG (IC_50_ = 0.56 ± 0.52 μg/mL) (Table 2).

### 2.5. 5α-Reductase Inhibitory Activity

Western blotting conducted to evaluate 5α-reductase type 1inhibitory activity in HaCaT cell. 5α-reductase type 1 inhibitory activity of QML showed potent activities compared with the PC, dutasteride. (Figure 1a). And 5α-reductase type 1 inhibitory activity of PD showed great activities compared with the PC (Figure 1b).

## 3. Discussion

Inflammation triggers all types of AV. Hence, inhibition of inflammation is pathway central to anti-acne. NO, a relative free radical, is synthesized by inducible NO synthase (iNOS). NO production in acne patients can be damage to skin and cause AV [5,30,31]. In this study, QML and PD showed potent anti-inflammatory activity that QML and PD have an effect anti-acne. QML (IC_50_ = 1.45 ± 0.25 μg/mL) showed potent inhibitory activity on NO production compared with the PC, L-NMMA (IC_50_ = 0.55 ± 0.19 μg/mL). Also, PD (IC_50_ = 53.52 ± 9.34 µM) showed potent inhibitory activity on NO production compared with the PC, L-NMMA (IC_50_ = 14.81 ± 12.76 μg/mL). Thus, QML and PD have an effect on anti-AV result from inhibitory activity on NO production.

Activation of the Th17 cell that induced IL-6 secrete IL-17A and IL-17F which main effector cytokines in acne lesions. IL-17A and IL-17F activate neutrophils and can target different cell types such as keratinocytes, monocytes, fibroblasts. Therefore, IL-17A and IL-17F lead to epidermal hyperplasia on keratinocytes. In addition, IL-17A and IL-17F stimulate keratinocytes and activate vascular inflammation [8,9,32,33]. Production of IL-8 encourages destruction of keratinocytes by TLR2 and TLR4 signal. And IL-8 performs initiation of inflammatory cytokines due to production of reactive oxygen species (ROS). ROS damages to enzyme, DNA and keratinocytes In addition, ROS stimulates production of inflammatory cytokines [2,34,35]. In this study, QML and PD showed more potent anti-inflammatory activity on cytokine production. In IL-6, QML (IC_50_ = 9.37 ± 1.50 μg/mL) showed more potent anti-inflammatory activities compared with the PC, EGCG (IC_50_ = 2.98 ± 1.47 μg/mL). PD (IC_50_ = 6.59 ± 1.66 µM) displayed stronger anti-inflammatory activities than EGCG (IC_50_ = 6.68 ± 1.86 μg/mL). In IL-8, QML (IC_50_ = 6.38 ± 2.58 μg/mL) showed more potent anti-inflammatory activities compared with the PC, EGCG (IC_50_ = 0.74 ± 0.09 μg/mL). PD (IC_50_ = 0.09 ± 0.41 µM) displayed stronger anti-inflammatory activities than EGCG (IC_50_ = 0.56 ± 0.52 μg/mL). Finally, QML and PD have an effect on anti-AV result from anti-inflammatory activity on cytokine production.

5α-reductase has isoenzymes which 5α-reductase type 1 and type 2. In particular, 5α-reductase type 1 is related to sebum production and causes of AV. 5α-reductase type 1 has activity in sebaceous gland. Hence, sebum production is associated with DHT which changes from testosterone by 5α-reductase type 1. Therefore, 5α-reductase type 1 inhibitory activity decreases DHT that decreases sebum production and AV [18,36,37]. In this study, QML and PD showed great 5α-reductase type 1 inhibitory activity compare with the PC. Hence, these results showed QML and PD reduced AV.

## 4. Materials and Methods

### 4.1. Plant Material

The leaves of QM were collected from Yeoju Eco Park, Yeoju, Republic of Korea (GPS coordinates: 37.346648, 127.494519) in July 2018. The plant was identified by Dr. Sung Sik Kim (Korea National Arboretum). A voucher specimen was sited at the herbarium of the College of Pharmacy, Chung-Ang University.

### 4.2. Chemical and Reagents

Dulbecco’s Modified Eagle Medium (DMEM), fetal bovine serum (FBS) and trypsin were purchased from Welgene (Gyeongsan, Republic of Korea). Streptomycin-penicillin was purchased from Gibco (NY, USA). Calcium-free DMEM, superscript^TM^ Ⅳ first-stand synthesis system and dream taq Green PCR Mix were purchased from Thermo Fisher Scientific (MA, USA). Sodium dodecyl sulfide (SDS), dithiothreitol (DTT), agarose, lipopolysaccharide (LPS), phosphate-buffered saline (PBS), 1,1-Diphenyl-2-picrylhydrazyl (DPPH) and Griess reagent (0.1% naphthylethylenediamine and 1% sulfanilamide in 5% H_3_PO_4_ solution), NG-Methyl-L-arginine acetate salt (L-NMMA) and thiazolyl blue tetrazolium bromide (MTT) and Dutasteride were purchased from Sigma Aldrich (St. Louis, USA). Reagent set B, cytokine IL-6, IL-8 ELISA sets used for immunoassay were purchased from BD Biosciences (NJ, USA). The primary and secondary antibodies for 5α-reductase type 1 were purchased form Abcam (Cambridge, UK). PVDF membrane and Mini-protean precast gels and were purchased form Bio-Rad (Hercules, CA, USA). ECL detection reagent (GE Healthcare Life Science, NJ, USA).

### 4.3. Cell Culture

RAW 264.7 cells were purchased from the Korean Cell Line Bank. HaCaT cells (Human, Adult, low Calcium, High Temperature, epithelial keratinocyte cell line) were purchased form the Dermatology of Chung-Ang university hospital. These cells were grown at 37 °C in a humidified atmosphere (approximately 5% CO_2_) in a DMEM medium containing 10% FBS, 100 IU/mL penicillin G and 100 mg/mL streptomycin.

### 4.4. MTT Assay

Before the biologic assay, the cytotoxicity was measured by the mitochondrial-dependent reduction of MTT to formazan. After cells were seeded in a 96-well plate and incubated for 6 h, the cells were treated with the samples (12.5, 25, 50 and 100 μg/mL or μM). The cells were incubated for an additional 24 h and the medium was replaced with phosphate buffered saline (PBS) containing 0.5 mg/mL MTT and the incubation continued for a further 4 h at 37 °C. Then the PBS was removed, and the MTT-formazan was dissolved in 100 μL of dimethyl sulfide. The extent of the reduction of MTT to formazan within the cells was quantified by measuring the absorbance at 540 nm using an ELISA reader. The cytotoxicity was calculated as cell viability (%) = sample O.D. / blank O.D. × 100.

### 4.5. Measurement of Inhibitory Activity on NO Production

RAW 264.7 macrophage cells were seeded in a 96-well plate and incubated for 6 h at 37 °C in humidified atmosphere (approximately 5% CO_2_). The cells were incubated in a serum free medium containing each sample and 1 μg/mL LPS. After incubating for an additional 20 h, the NO contents were evaluated by Griess assay. After getting supernatant from the cells treated with the samples, the Griess reagent was added. L-NMMA was used as a positive control. Inhibitory activity on NO production was calculated as the rate of inhibition (%) = 100 − (sample O.D. − blank O.D.) / (control O.D. − blank O.D.) × 100. IC_50_ values were defined as the concentration that inhibit 50% of NO production.

### 4.6. Measurement of Inhibitory Activity on Cytokine Production

HaCaT cells were seeded in a 96-well plate and incubated for 6 h at 37 °C in humidified atmosphere (approximately 5% CO_2_). The cells were incubated in a medium containing 1 μg/mL LPS. After incubating for an additional 24 h, the supernatant from the cells were treated with the samples used for the assay using ELISA kit. Cytokine contents were quantified by measuring the absorbance at 450 nm using an ELISA reader.

### 4.7. Western Blot Assay

HaCaT cells (3 × 10^5^ cells/mL) were pre-incubated for 16 h and then treated QML and PD. After incubation for 24 h, the cells washed with PBS and the cell lysates were prepared with RIPA buffer. The cell lysates were centrifuged at 13,000 × rpm for 15 min at 4 °C. The cell lysates were subjected to electrophoresis in SDS-polyacrylamide gels (10%) and the separated proteins were transferred on to a PVDF membrane. The membrane was blocked with blocking buffer for 60 min at room temperature. Then the membrane was probed with primary antibody for 5α-reductase for overnight at 4 °C. After washing, the blots were incubated with secondary antibody for 1 h at room temperature. The bands were visualized by LAS-4000 luminescent image analyzer (GE Healthcare Life Science, NJ, USA) using ECL detection reagent.

## 5. Conclusions

In this study, QML and PD showed potent anti-inflammatory activity. In additional QML and PD showed good 5α-reductase inhibitory activity. Hence, these results showed that QML and PD may have an anti-AV effect.

## Figures and Tables

**Figure 1 molecules-25-02154-f001:**
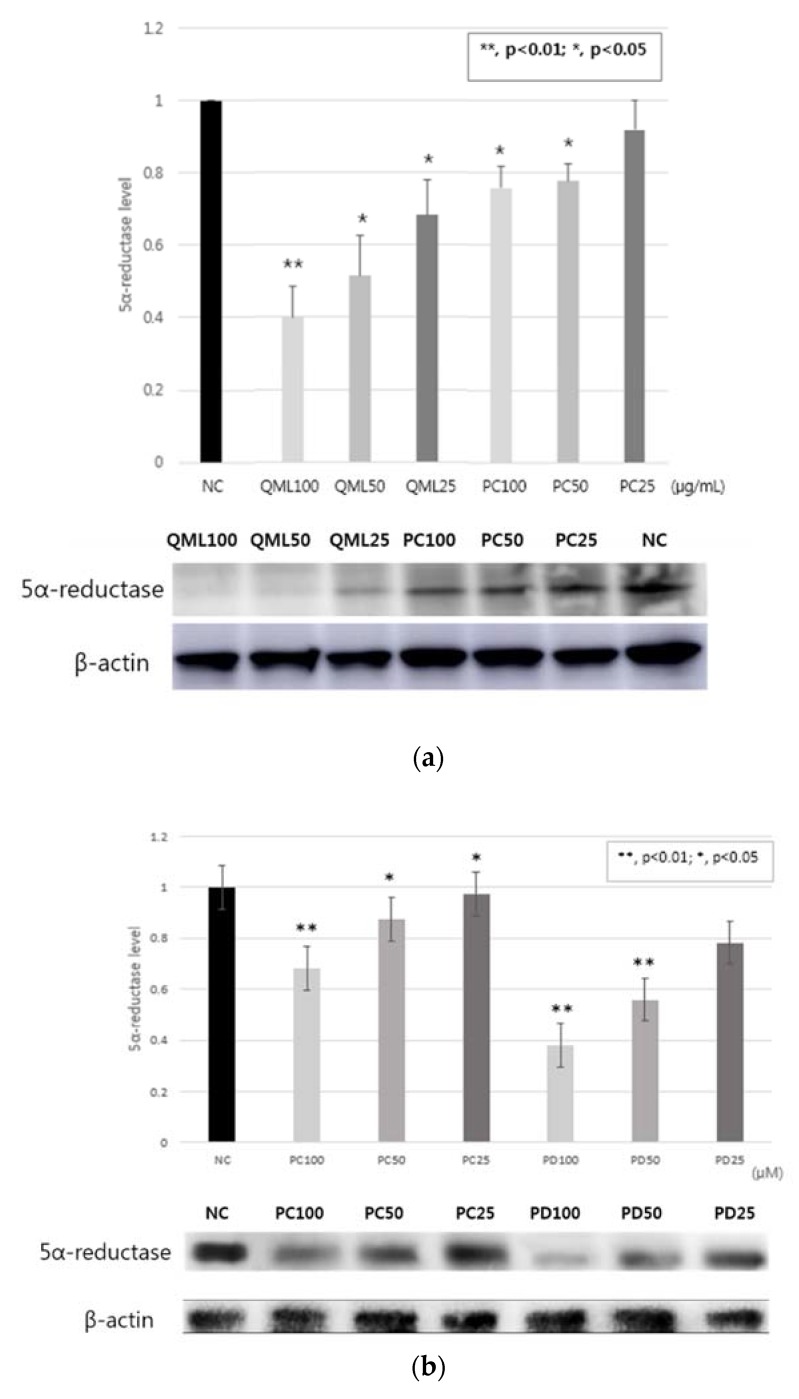
5α-reductase type 1 inhibitory activity in HaCaT cell. (**a**) QML; (**b**) PD. Data are expressed as the means ± standard deviation (SD) of three determinations. * *p* < 0.05; ** *p* < 0.01, compared with normal control. NC: normal control; PC: positive control.

**Table 1 molecules-25-02154-t001:** IC_50_ values of *Quercus mongolica* leaf extract (QML) and pedunculagin (PD) on inhibitory activity of nitric oxide (NO) production.

Samples	NO Production IC_50_ (μg/mL)	NO Production IC_50_ (µM)
QML	1.45 ± 0.25 ^a^	–
PD	–	53.52 ± 9.34 ^b^
L-NMMA	0.55 ± 0.19 ^b^	14.81 ± 12.76 ^a^

Values represent means ± standard deviation (SD) of three determinations. a-b: in the same columns are significantly different (*p* < 0.05).

**Table 2 molecules-25-02154-t002:** IC_50_ values of QML and PD against inhibitory activity on IL-6, IL-8 production.

Samples	IL-6 IC_50_ (μg/mL)	IL-6 IC_50_ (µM)	IL-8 IC_50_ (μg/mL)	IL-8 IC_50_ (µM)
QML	9.37 ± 1.50 ^a^	–	6.38 ± 2.58 ^a^	–
PD	–	6.59 ± 1.66 ^b^	–	0.09 ± 0.41 ^b^
EGCG	2.98 ± 1.47 ^b^	6.68 ± 1.86 ^a^	0.74 ± 0.09 ^b^	0.56 ± 0.52 ^a^

Values represent means ± standard deviation (SD) of three determinations. a-b: in the same columns are significantly different (*p* < 0.05).

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
