# Peer review of "Anti-Acne Vulgaris Effects of Pedunculagin from the Leaves of Quercus mongolica by Anti-Inflammatory Activity and 5α-Reductase Inhibition"

_molecules, 2020, doi:10.3390/molecules25092154_

Round 1
Reviewer 1 Report
This is an interesting manuscript and publication of the current manuscript is recommended only after proofreading.
Author Response
Response to Reviewer 1 Comments
This is an interesting manuscript and publication of the current manuscript is recommended only after proofreading.
Answer: The manuscript completed proofreading.
Reviewer 2 Report
My remarks are presented in the attached file.

Author Response
We wrote long anwer to your comments. Please see the attachment.

Reviewer 3 Report
Dear Authors
Very interesting research and results. But you need to improve:
line 167 - please provide GPS coordinates
Fig 2b - the image is of poor quality
- all references should have DOI
References 10, 20, 24, 32, 33 are too old and should be removed or replaced by new current reports
Author Response
Response to Reviewer 3 Comments
Dear Authors
Very interesting research and results. But you need to improve:
line 167 – please provide GPS coordinates
Fig 2b – the image is of poor quality
- all references should have DOI
References 10, 20, 24, 32, 33 are too old and should be removed or replaced by new current reports
Answer:
- GPS coordinates are 37.346648, 127.494519. We add GPS coordinates in manuscript. (Lines 134-135)
- Fig 2b eliminated in manuscript. Because we discovered wrong concept of filaggrin and cornified envelope (CE) formation. So we decided to eliminate result of filaggrin and CE formations with the related figures.
- We checked all references DOI once again.
- We removed references 10, 20, 24, 32, 33.
Round 2
Reviewer 2 Report
Please see the remarks in the attached file.

Author Response
Response to Reviewer 2 Comments - round 2
The authors provided an improved version of their paper. However, there are still some remarks that I would like to underline:
- Abstract: authors should provide in abstract, shortly, the results of their work.
Answer: We changed abstract as below.
“Quercus mongolica (QM), a member of the Fagaceae family, has been used as traditional medicine in Korea, China, and Mongolia as a treatment for inflammation of oral, genital or anal mucosa, and for external inflammation of skin. To treat acne vulgaris (AV), we evaluated the inhibition of inflammatory cytokines (IL-6 and IL-8) of QM leaves extract (QML) and its main compound, pedunculagin (PD) in vitro and 5α-reductase inhibitory activity by western blotting. As results, QML and PD showed potent NO production inhibitory activity compared with the positive control (PC), NG-monomethyl-L-arginine (L-NMMA). QML and PD was also showed the decreases of IL-6 and IL-8 compared with the PC, EGCG and exhibited potent 5α-reductase type 1 inhibitory activities compared with the PC, dutasteride.” (Lines 11-19)
- Line 31-32: Free radicals and nitric oxide are not inflammatory responses. Please reformulate. Also, reference 1 does not support the affirmations that were made (as far as it can be seen from abstract, as the article is not provided in a language written with Latin alphabet). Please change reference.
Authors failed to explain the link between nitric oxide production and acne. Providing more data on this topic they may better justify their experiments and so they may suggest a possible link to acne vulgaris.
Answer: We changed manuscript as below.
“Keratinocytes play a role in skin barrier function. When these keratinocytes are exposed to external stimuli like air pollution, stress and ultraviolet radiations it can induce mediator of inflammation, nitric oxide (NO) and inflammatory response such as inflammatory cytokines. Hence, NO and the inflammatory cytokines might cause acne vulgaris (AV) [1-4].” (Lines 23-26)
And we deleted reference 1 and added another references (1 and 2).
And we explained the link between free radicals, nitric oxide production and acne as below.
“NO is a free radical that damages to cell, tissue and DNA. And NO can form peroxynitrite that is a potent cytotoxic agent damage to molecules in cells, including DNA. These damages evoke many chronic inflammatory diseases such as psoriasis and pathogenesis of AV. It also was reported that nitrate and nitrite levels which are an index of NO production were high in acne patients [5, 6].” (Lines 27-30)
- Line 36: authors should link the modifications induced by cytokines to acne vulgaris (AV) rather than to skin barrier disruption. Please reformulate.
Answer: We changed introduction and discussion as below.
“Particularly, IL-6 induce differentiation of Th17 cells. Activation of the Th17 cell produce IL-17 that the cytokine contributes significantly to the inflammatory response in AV. Also, IL-17 found in acne lesions [8-10]. (Lines 32-34)
“Activation of the Th17 cell that induced IL-6 secrete IL-17A and IL-17F which main effector cytokines in acne lesions. IL-17A and IL-17F activate neutrophils and can target different cell types such as keratinocytes, monocytes, fibroblasts. Therefore, IL-17A and IL-17F lead to epidermal hyperplasia on keratinocytes. Also, IL-17A and IL-17F stimulate keratinocytes and activate vascular inflammation [8, 9, 32, 33].” (Lines 124-128)
- Line 38: „Acne vulgaris (AV) is a chronic inflammatory disease caused by sebum production”. Normal sebum production does not produce acne. Please reformulate.
Answer: We changed sentence as below and added reference 13, 14.
“AV is a chronic inflammatory disease caused by excessive sebum production [13, 14].” (Line 36)
- Lines 42-43: „ Keratinocytes became hyperkeratinization increase sebum production in sebaceous gland. These DHT convert from testosterone by 5α-reductase reactions”. Please reformulate, because of the words marked in red, the sentences are not clear.
Answer: We changed the sentence in introduction as below.
“If abnormal hyperkeratinization occur in keratinocytes, sebum production increases excessively in sebaceous gland [15-17]. 5α-reductase converts DHT from testosterone.” (Lines 39-41)
- Lines 55-57: In this study authors didn’t prove an improvement of AV. They studied the anti-inflammatory activity and the reduction of 5α reductase activity, which may improve acne, but no certitude exists until tests on acne patients. Authors, authors should reformulate the objectives of the study according to what they actually did.
Answer: We conducted this study in vitro. So we added word ‘in vitro’ in abstract. Further study, we will test on acne patients.
- Line 149: „ NO, a relative free radical, synthesized by inducible NO synthase (iNOS):” This is a fragment, no a sentence. Please reformulate.
Answer: We changed the sentence as below.
“NO, a relative free radical, is synthesized by inducible NO synthase (iNOS).” (Line 117)